# Effect of Cooling Rate on the Crystal Quality and Crystallization Rate of SiC during Rapid Solidification Based on the Solid–Liquid Model

**Xiaotian Guo** [1,*], **Yue Gao** [2], **Zihao Meng** [1] **and Tinghong Gao** [2,*]

1   School of Physics and Electronic Engineering, Xinxiang University, Xinxiang 453003, China; zihaomeng2022@163.com
2   Institute of Advanced Type Optoelectronic Materials and Technology, College of Big Data and Information Engineering, Guizhou University, Guiyang 550025, China; gs.ygao20@gzu.edu.cn
*   Correspondence: guoxiaotian@xxu.edu.cn (X.G.); thgao@gzu.edu.cn (T.G.)

**Abstract:** The silicon carbide (SiC) that can achieve better electron concentration and motion control is more suitable for the production of high temperature, high frequency, radiation resistance, and high-power electronic devices. However, the fabrication of the high purity single crystal is challenging, and it is hard to observe the structural details during crystallization. Here, we demonstrate a study of the crystallization of single-crystal SiC by the molecular dynamic simulations. Based on several structure analysis methods, the transition of the solid–liquid SiC interface from a liquid to a zinc-blende structure is theoretically investigated. The results indicate that most of the atoms in the solid–liquid interface begin to crystallize with rapid solidification at low cooling rates, while crystallization does not occur in the system at high cooling rates. As the quenching progresses, the number of system defects decreases, and the distribution is more concentrated in the solid–liquid interface. A maximum crystallization rate is observed for a cooling rate of $10^{10}$ K/s. Moreover, when a stronger crystallization effect is observed, the energy is lower, and the system is more stable.

**Keywords:** silicon carbide; solid–liquid model; structural evolution; crystallization



## 1. Introduction

Silicon carbide (SiC), a wide-band gap semiconductor, has unique properties such as high strength and stiffness, effective oxidation resistance, excellent corrosion resistance, high thermal conductivity and stability, and high electron velocity [1]. At present, more than 250 different polytypes of SiC have been observed, with 3C-SiC (zinc-blende) and 4H-SiC (wurtzite) being the most common. The 3C, 4H, and 6H polytypes exhibit different stacking orders; 3C has three diatomic layers (ABC), 4H exhibits four diatomic layers (ABAC), and 6H has six biatomic layers that are periodically stacked (ABCACB) [2,3]. The band gap of cubic (3C)-SiC is 2.36 eV, and the structure of zinc-blende has the advantages of a high melting point, high thermal conductivity, and high critical breakdown electric-field [4].

Melting and solidification are the two important processes involved in the preparation of single crystals by using the liquid phase method. Meanwhile, an in-depth understanding of the physics of solid–liquid coexistence is a key factor in determining the mechanisms that control the microstructural characteristics of a material [5,6]. Research on the liquid–solid transformation and the microstructural evolution during solidification has received substantial interest in condensed matter physics and material science [7]. Molecular dynamics (MD) simulations [8,9] can provide detailed atomic-scale structural information regarding the solidification process. Thus, MD simulations are widely used to study the crystallization and amorphization of liquid metals and alloys, as well as semiconductor materials [10–13]. It is difficult to prepare large-size silicon carbide in experiments, and

a solid–liquid coexistence state can exist in the local range during the thermal treatment. Therefore, we constructed a solid–liquid model to perform molecular dynamics simulations and wanted to investigate the microstructure evolution during the induced crystallization process. Recently, a new method for the synthesis of SiC nanoparticles, pulsed laser ablation in liquid (PLAL), has been proposed. The PLAL method uses pulsed laser ablation of the target to synthesize nanoparticles, which forms a liquid–solid interface and generates high temperatures when a laser beam with sufficient energy is irradiated onto the target [14–16]. Moreover, MD simulation methods can be used to assess the microstructural evolution of a material during solidification; however, corresponding experiments are not available. By simulating the solid–liquid model of SiC solidification process, the mechanism of crystal growth can be explored. Research on the short-range order of amorphous SiC [17] and liquid SiC [18] has been correctly carried out using MD. However, only a few MD simulation reports on the melting of SiC are available. Zhou et al. [19]. compared the bulk melting, surface melting, and crystal growth of SiC using the Tersoff potential and the modified embedded atom method potential.

In this paper, the MD method with the commonly used Tersoff potential [18] was used to study the microstructural evolution of the solid–liquid interface of 3C-SiC during rapid solidification processes. These MD simulations provide us with a better understanding of the crystallization rate, crystal quality, and defect distribution during rapid quenching by radial distribution function (RDF), angle bond angle function, and Voronoi polyhedron index.

## 2. Simulation Conditions and Methods

The conditions for the simulation calculations are as follows. A total of 64,000 SiC atoms are arranged in a cubic box, and the system runs under periodic boundary conditions. For the solid–liquid phase model, 32,000 atoms on the left side have a zinc-blende crystal structure, while the remaining atoms on the right side of the simulation box have a disordered structure.

Figure 1 shows the section of the SiC S/L phase model in the middle position. The atomic coordinates of the liquid region were obtained from completely melted SiC at 5500 K. The present work is based on a constant-temperature, constant-pressure MD (NPT MD [20]) approach, and the atom interactions are expressed by Tersoff potentials [21]. It is a widely used equation in various applications for silicon, carbon, germanium, and their compounds, to simulate a covalent system with complex structure and energy. The potential can be written as:

$$E = \sum_i E_i = \frac{1}{2} \sum_{i \neq j} V_{ij} \tag{1}$$

$$V_{ij} = f_c\left(r_{ij}\right)\left[f_R\left(r_{ij}\right) + b_{ij} f_A\left(r_{ij}\right)\right] \tag{2}$$

where the potential energy is decomposed into site energy $E_i$ and a bonding energy $V_{ij}$; $r_{ij}$ is the distance between the atoms $i$ and $j$; $f_A$ and $f_R$ are the attractive and repulsive pair potentials, respectively; and $f_c$ is a smooth cut-off function. More details can be found elsewhere [21].

The software program used in this simulation is LAMMPS [22]. Through an improved method proposed by Yoo [23] to build the solid–liquid model of SiC, Yan [24] have calculated the equilibrium melting point (3112 K) based on the Tersoff potential. This temperature is close to the experimental SiC melting point of 3103 K [25]. The simulation time step applied was 1 fs ($1 \times 10^{-15}$ s), and the NPT MD simulation is started at 3500 K, which is approximately 400 K higher than the standard melting temperature of SiC. After a relaxation time of 1 ns, the concave and convex solid–liquid interface is obtained. The system is quenched to 200 K at four cooling rates, i.e., R1: $10^{10}$ K/s, R2: $10^{11}$ K/s, R3: $10^{12}$ K/s, and R4: $10^{13}$ K/s, which are achieved by controlling the kinetic rate of the system. During each quenching process, the atomic volume and the total energy per atom are calculated at 5-K intervals, and the structural configurations are recorded at 5-K intervals

during quenching from 3500 K to 200 K. Finally, the RDF [26], crystallization speed, and visualization technology were used to examine the evolution of the crystal structures and the microstructural properties of the SiC solid–liquid interface during rapid cooling.

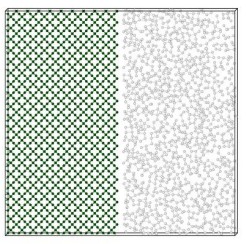 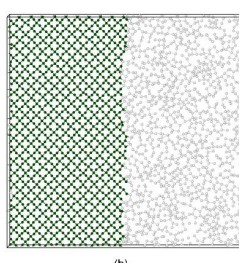 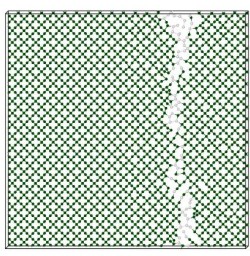

**Figure 1.** Section of the SiC S/L phase model in the middle position. The Miller index of S/L interface is (1 0 0). (The green atoms are solid atoms and the gray atoms are liquid atoms.) (**a**) The initial S/L phase model. (**b**) The configuration of stress-relieving. (**c**) The system is quenched to 200 K at R1 cooling rate.

## 3. Results and Discussion

### 3.1. Radial Distribution Function

The RDF is a useful tool for analyzing the structural characteristics of liquid and amorphous structures. Because the left region of the solid–liquid model includes 32,000 atoms with a zinc-blende crystal structure, denoting few structural changes, the structural properties of the atoms in the right region of the solid–liquid model, which have disordered initial configurations, can be traced during the rapid cooling process based on the RDF.

Figure 2 compares our calculated RDF and PRDF (Partial radial distribution function) results. The first peak in g(r) for the SiC crystal at 1.9 Å is attributed to Si-C bonds, indicating that these bonds in the SiC crystal have an average length of 1.9 Å, corresponding to a zinc-blende structure. When the temperature exceeds the melting point, the average C-C bond length is 1.5 Å, the average Si-C bond length is 1.9 Å, and the average Si-Si bond length is 2.4 Å. This result indicates that after melting, the shortest bond in SiC is the C-C bond, and the longest bond is the Si-Si bond. When SiC melts, the $g_{si-c}(r)$ has a peak distance of 1.9 Å from the first nearest neighbor and 3.0 Å from the second nearest neighbor, consistent with the bond length in the crystal. The peak distance for the first nearest neighbor of the $g_{C-C}(r)$ is near 1.5 Å, and that of its second nearest neighbor is near 2.6 Å, which is consistent with simulation results for liquid C [27]. The peak distance of the first nearest neighbor for the Si-Si bond is 2.4 Å, and that of the second nearest neighbor is approximately 3.2 Å, which is not substantially different from simulation results for liquid Si [28].

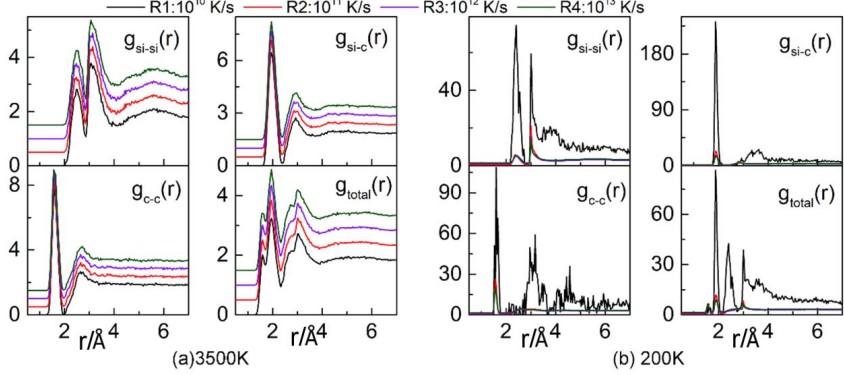

**Figure 2.** Evolution of the RDF of the liquid phase (the liquid region on the right) for different cooling rates at 3500 K (**a**) and 200 K (**b**).

After the system has been quenched and has reached a temperature of 200 K, $g_{Total}(r)$ and the partial RDFs of $g_{C-C}(r)$, $g_{C-Si}(r)$, and $g_{Si-Si}(r)$ show a clear crystallization trend for a cooling rate of $10^{10}$ K/s. However, no crystallization is observed at higher cooling rates.

### 3.2. Crystallization Rate

The crystallization rate reflects the rate at which a substance changes from the liquid state to the crystalline state. The evolution of crystalline structures in SiC was tracked during the quenching process, and the crystallization rate was calculated as the system was cooled from 3500 K to 200 K in intervals of 5 K, as depicted in Figure 3. The Y-coordinate represents the increments in cubic diamond during the cooling process for intervals of 5 K. We can observe that the crystallization rate is highest for a cooling rate of $10^{10}$ K/s, particularly at the initial temperature. As the temperature decreases, the crystallization rate decreases significantly and then increases. After several transitions, the crystallization rate reaches a minimum, oscillates about 0, and then flattens out. As the cooling rate increases, the crystallization rate oscillates around 0, demonstrating that the system does not crystallize at high cooling rates; instead, the system is in a state of dynamic equilibrium. Only at low cooling rates (on the order of $10^{10}$ K/s or lower) does the crystal continue to crystallize.

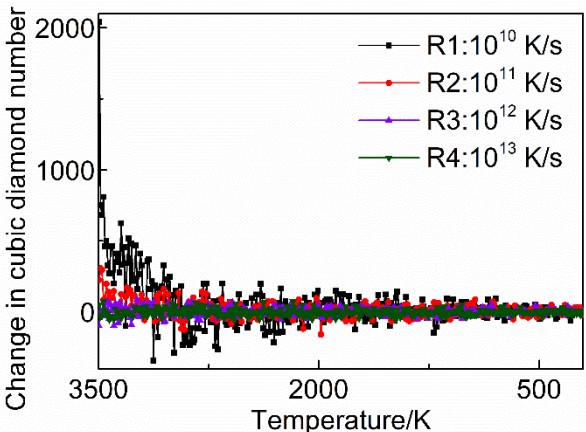

**Figure 3.** Crystallization rates for different cooling rates.

### 3.3. Bond Angle Analysis

The bond angle provides a useful indicator for assessing the relationships among neighboring atoms. The bond angle distribution is calculated from the first minimum value of g(r).

The curve presented in Figure 4 indicates changes in the bond angle distribution width during the quenching process for different cooling rates at 3500 K (left) and 200 K (right). The results demonstrate that the bond angle distribution varies from 40° to 180° at 3500 K, with a main peak at 109°28′. At 200 K, the bond angles are concentrated between 80° and 160°. The existence of bond angles near 109° indicates the presence of some nonstandard tetrahedron units or other complex local structures beyond the abundant three-fold coordinated atoms in the system. For a cooling rate of $10^{10}$ K/s, due to the high crystallization degree of the system, the bond angle distribution function is sharper, with most of the angles between 100° and 110°, indicating that the zinc-blende crystal structure is dominant at this cooling rate. However, as the cooling rate increases, the crystallization ability decreases, and the bond angle distribution is flat; meanwhile, we can observed that lower cooling rates correspond to higher peak values.

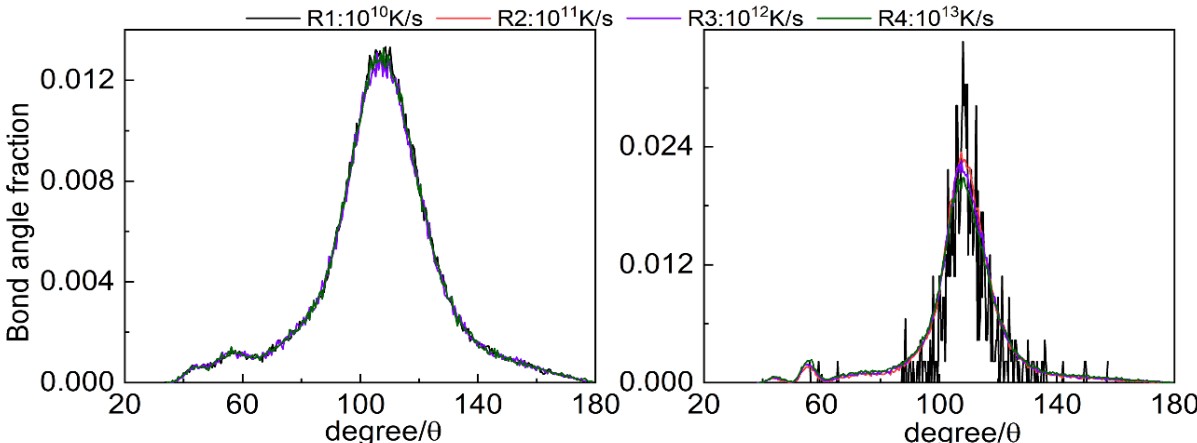

**Figure 4.** Bond angle distribution function for different cooling rates at 3500 K and 200 K.

*3.4. Energy and Mean Square Displacement (MSD) Analysis*

The relation between the average atomic potential energy and temperature during quenching is shown in Figure 5. The potential energy is an important parameter to consider when assessing the stability of a simulation system. This parameter directly reflects many important structural changes, such as melting and crystallization. During the quenching process, no obvious jumps in the energy curves are observed for the SiC solid–liquid interface at cooling rates of $10^{11}$ K/s to $10^{13}$ K/s, as shown in Figure 5. The energy curves are linear at both high and low temperatures, as expected for typical liquid and solid states. When the cooling rate is $10^{10}$ K/s, the energy distribution function has an obvious turning point at 3300 K.

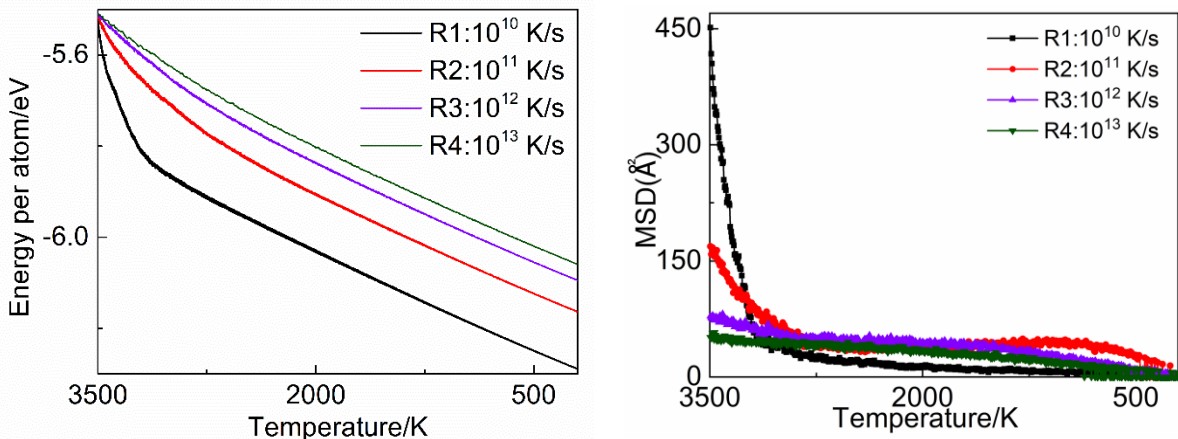

**Figure 5.** Variation in the average potential energy per atom as a function of temperature for different cooling rates. (**left**) MSD of four different cooling rates. (**right**).

According to the laws of thermodynamics, lower energies correspond to more stable systems. Obviously, as the temperature continuously decreases, the free energy in the solid state remains lower than that for the liquid state. Compared with liquids, solids are more stable, which leads to the solidification of melted SiC. However, if the cooling rate is too high, the melt will directly transform from a liquid to an amorphous structure, as there is not sufficient time for the atoms to be rearranged. Here, the critical cooling rate for SiC solidification is $10^{10}$ K/s.

The mean square displacement (MSD) can be used to describe the migration of atoms. It can be observed from the MSD curve that at the beginning of quenching, the lower the cooling rate is, the more it decreases. After 3300 K, MSD gradually drops smoothly to 0. In the solidification process, it is indicated that the diffusion of atoms progresses rapidly until

it becomes stable. The cooling rate has a great influence on the diffusion of atoms, and the lower the cooling rate, the more intense the diffusion between atoms.

### 3.5. Voronoi Analysis

Proposed by Finney [29], the Voronoi polyhedron index, <n3 n4 n5 n6>, is a microstructural indicator that describes the local topological characteristics of amorphous structures. The variables n3, n4, n5, and n6 denote the number of trilateral, quadrilateral, pentagon, and hexagon structures, respectively, in the Voronoi polyhedron. For example, <4 0 0 0> represents a tetrahedral structure composed of four trilateral structures, such as crystalline Si or SiC. <2 3 0 0> corresponds to a pentahedron composed of two trilateral and three quadrilateral structures, and <2 3 0 0> represents a defect in <4 0 0 0> that is obtained by cutting out a corner of <4 0 0 0>, indicating that an additional neighboring atom exists in this direction [10,30].

Figure 6 shows that under different cooling rates, the final clusters formed by the system are mainly tetracoordinated (<4 0 0 0>), corresponding to 44,000 and 56,000 atoms, with a few <2 3 0 0> structures and very little <2 2 2 0> and <0 6 0 0> hexacoordination. The number of <4 0 0 0> structures increase with decreasing cooling rate, while the number of <2 3 0 0> structures decreases with reducing cooling rate. This result indicates that a lower cooling rate is conducive to the transformation of the high coordination structure to the <4 0 0 0> structure, giving the system more time for the atom positions to be adjusted.

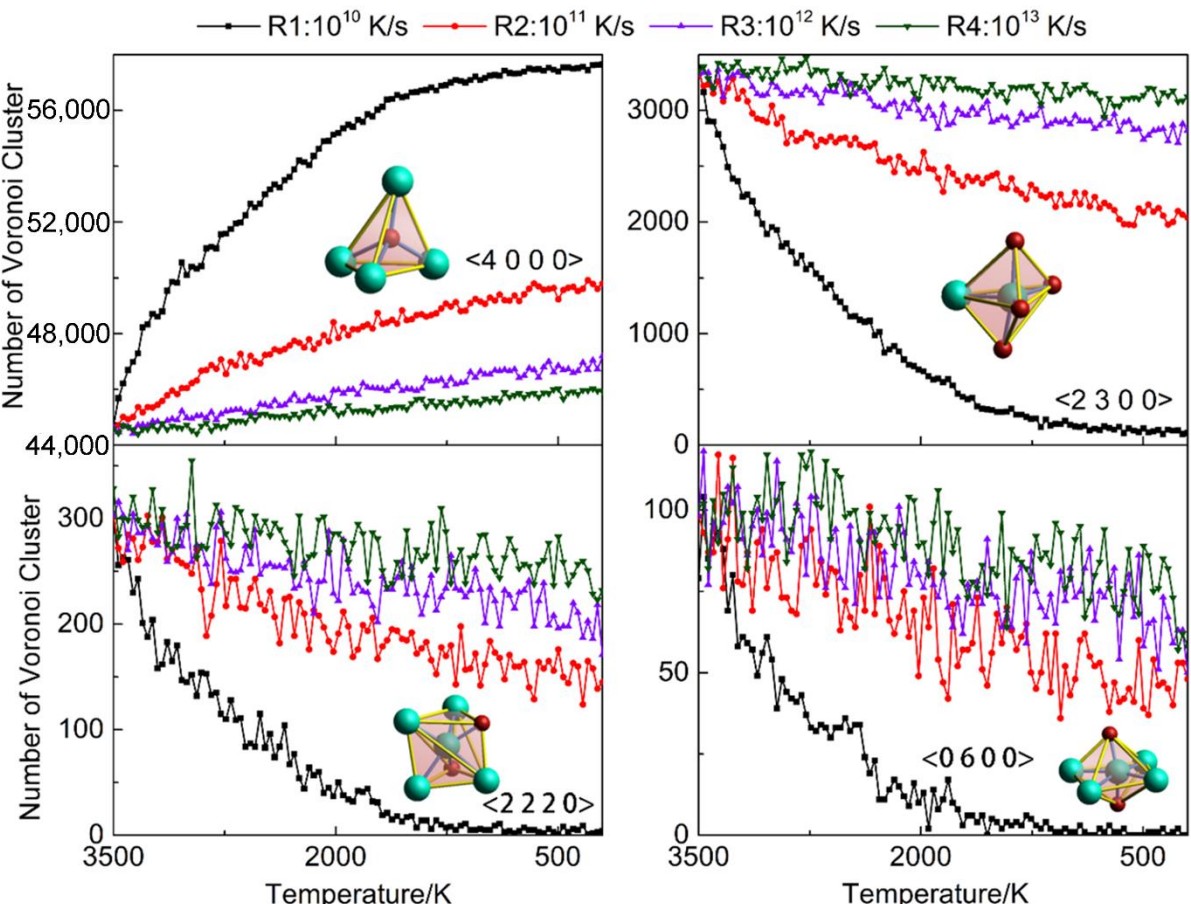

**Figure 6.** Variation in Voronoi polyhedron clusters as a function of temperature at four cooling rates.

### 3.6. Visualization

The shape and location of the solid–liquid interface and the defect distribution evolution at different cooling rates were investigated using visualization technology. The solid–liquid interface shape and position during the SiC crystal quenching process are

depicted in Figure 7 for 3500 K and 200 K. The first neighbor atoms of the zinc-blende crystalline region (green) and the liquid structures (gray) were analyzed using the dislocation analysis method [31–34].

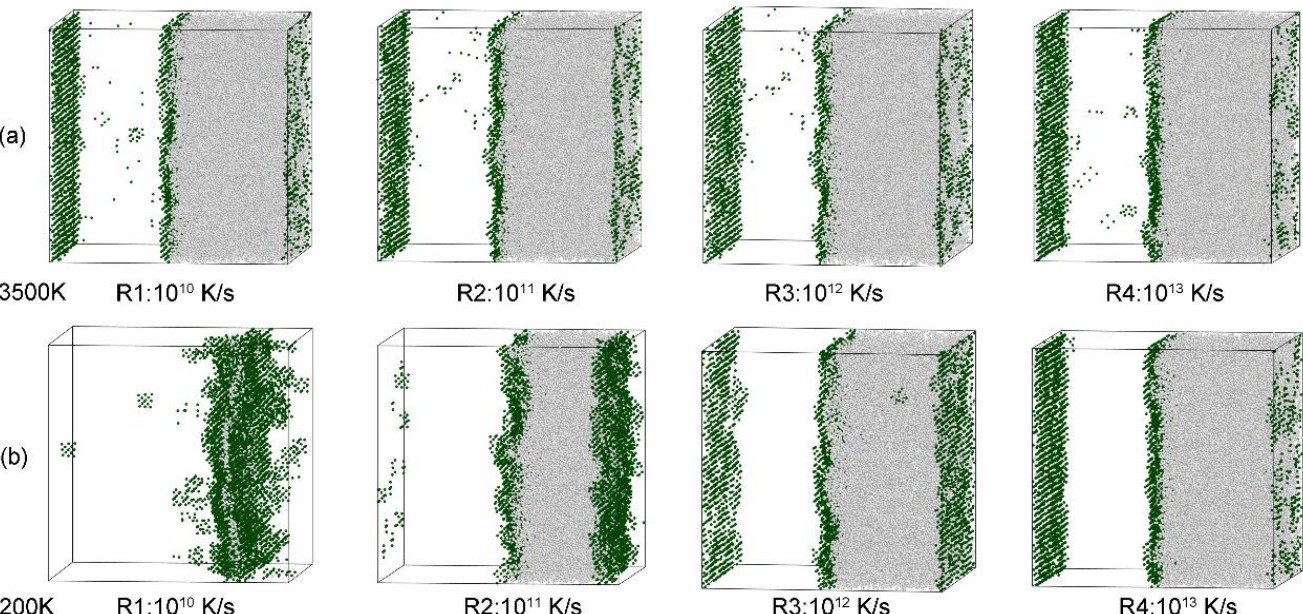

**Figure 7.** Evolution of the shape and location of the solid–liquid interface and defect distribution at different cooling rates. The green and gray spheres represent the first neighbor atoms of the zinc-blende crystal structure and the liquid structure, respectively.

In the quenching process, we can observe that the atoms at the solid–liquid interface have a greater range of motion; in contrast, the atoms in the solid region move very little. The atoms at the solid–liquid interface thus move towards the liquid atoms. The same conclusion can be obtained analytically, that is, a lower cooling rate corresponds to better crystallization. For a cooling rate of $10^{10}$ K/s, the region of liquid atoms is very small, almost invisible at the solid–liquid interface. For a cooling speed of $10^{11}$ K/s, the solid–liquid interface shows greater movement, and at $10^{12}$ K/s and $10^{13}$ K/s, the solid–liquid interface does not change. Moreover, the distribution of defects decreases with decreasing temperature, with the defects being focused at the solid–liquid interface. As the temperature decreases, the number of defects decreases, revealing that the degree of short-range order in the system increases.

### 3.7. Defect Statistics

Figure 8 shows statistical results regarding the numbers of different atom types for different cooling rates. The abscissa is the temperature, and the ordinate is the number of atoms. The black, red, purple, and green lines correspond to cooling rates of $10^{10}$, $10^{11}$, $10^{12}$, and $10^{13}$ K/s, respectively.

As we all know, defects destroy the integrity of the internal structure in crystal, and thus destroy the symmetry of the crystal. Crystalline SiC has a cubic diamond structure, and defect positions correspond to the first and second neighbors of the cubic diamond, where the structures are often distorted. During quenching, the defects transition from a liquid atom to a second neighbor, to a first neighbor, and then to a cubic diamond atom, with continuing growth.

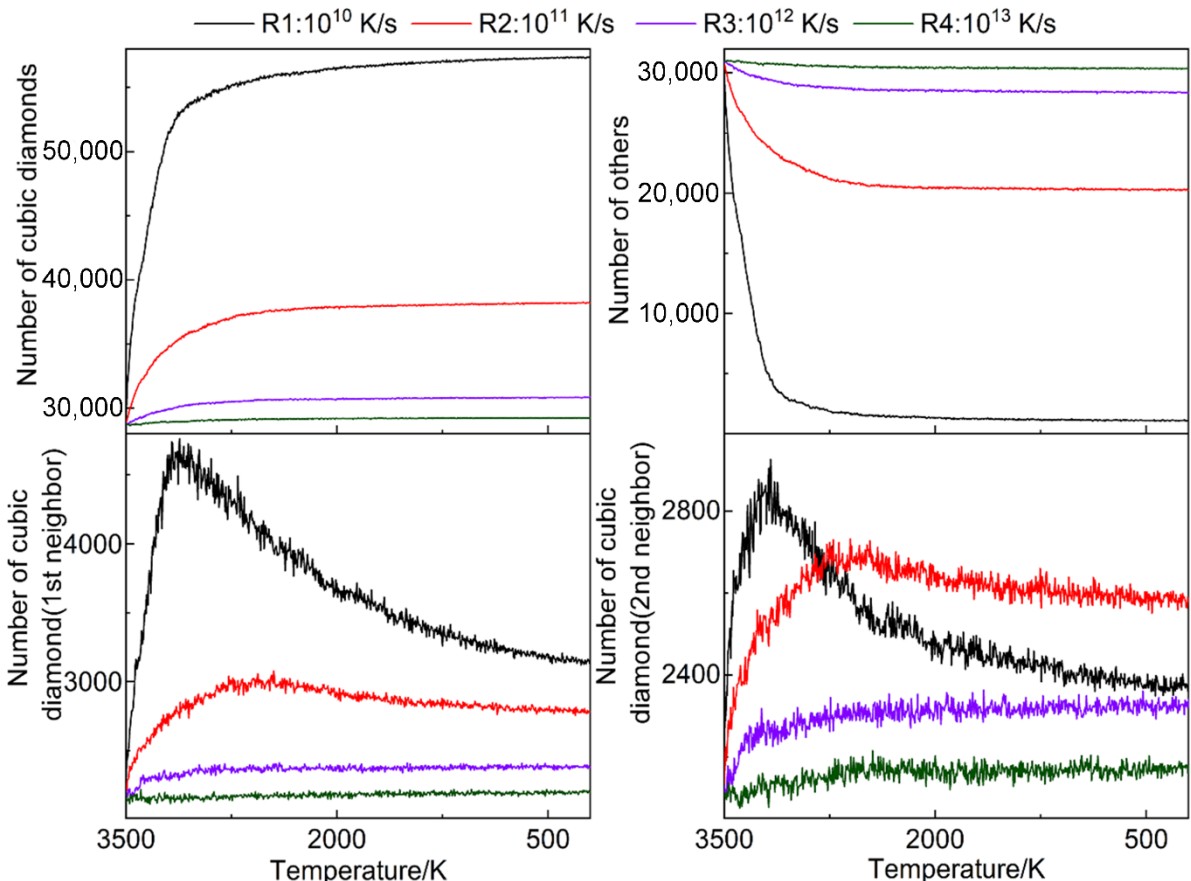

**Figure 8.** Statistical parameters for different cooling rates.

The trend for the number of cubic diamonds tends to be the opposite of the trends observed for the previous parameters. The number of first and second cubic diamond neighbors during the growth of the solid–liquid model was investigated. For a cooling rate of $10^{12}$ K/s or $10^{13}$ K/s, the overall trend is the same, and the values decrease slowly. For a cooling rate of $10^{10}$ K/s or $10^{11}$ K/s, the parameter first increases and then decreases. However, for a cooling rate of $10^{10}$ K/s, when the temperature reaches 3100 K, the curve exhibits a maximum. As the quenching time increases, the system continues to cool, and the degree of short-range order increases, with the system becoming more stable. Therefore, the number of defects continues to decrease. For a cooling rate of $10^{11}$ K/s, as the temperature decreases, the number of defects in the system initially increases and then decreases; however, when the temperature reaches 200 K, the number of defects is higher than that at the initial temperature. This result indicates that a lower cooling rate corresponds to a higher crystallization degree and crystal quality. However, for a high cooling rate of $10^{12}$ K/s or $10^{13}$ K/s, crystallization did not occur during quenching.

Figure 9 shows the position change of the solid–liquid interface, with the yellow area corresponding to the solid–liquid separation interface. The gray spheres denote liquid atoms. For a given temperature, as the cooling rate decreases, the solid–liquid interface becomes thinner, and the concave and convex degrees of the interface decrease. For a constant cooling rate, the thickness of the solid–liquid interface is consistent with the trend of the first and second cubic diamond neighbors. For a cooling rate of $10^{10}$ K/s, at a temperature of 3000 K, the number of defects is the largest, the interface is the thickest, and the concave and convex degrees are the highest. The same results are observed at 2800 K for a cooling rate of $10^{11}$ K/s. However, for cooling rates of $10^{12}$ K/s and $10^{13}$ K/s, due to the higher cooling speed, the atomic reaction at the solid–liquid interface was not sufficient, and the number of defects and the interface thickness did not change significantly.

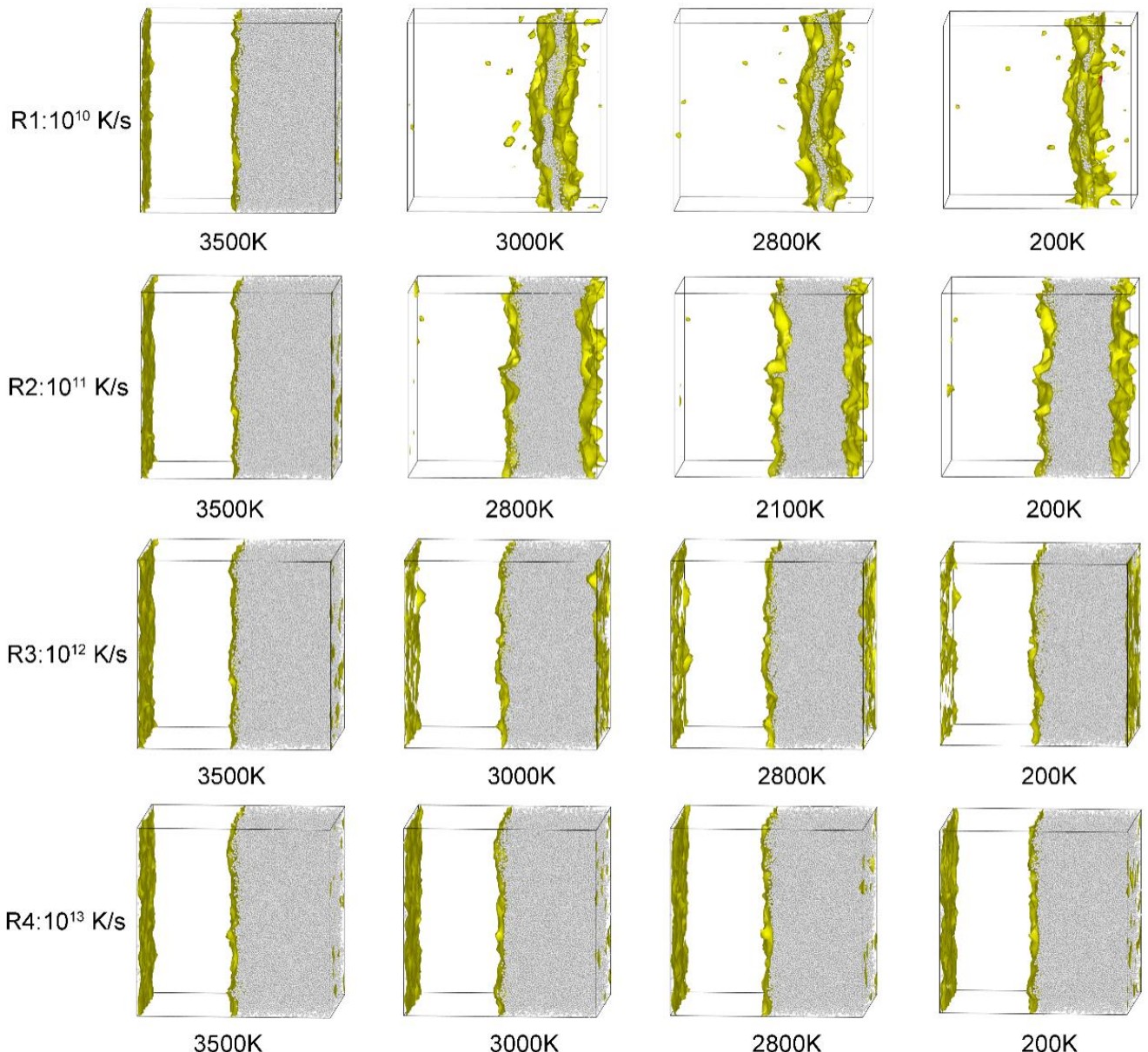

**Figure 9.** The position change of solid–liquid interface under different cooling rates (gray represents liquid atoms, and yellow shadows represent solid–liquid interface locations).

Figure 10 shows the change in section thickness of the solid–liquid interface. We can clearly see the interface changes; the lower the cooling rate, the more obvious the slice change. At cooling rate of R1: $10^{10}$ K/s, the interface section is thickest at 3000 K with a dramatic surface undulation in the S/L interface. This undulating surface increases the area of contact between solid and liquid, thus makes crystal easy to grow. With the decrease in temperature, the interface section becomes thinner and flat. At high cooling rates, the interface sections were always consistent with its initially undulating surface during quenching. The high cooling rates have a strong influence on crystallization rate by retarding the migration of S/L interface, which makes there be no obvious induced crystallization happening at a cooling rate beyond $10^{11}$ K/s. In some recent studies, the smoother the solid–liquid interface, the more difficult the crystal growth is. Therefore, it is considered that the solid–liquid interface energy becomes larger as the solid–liquid interface will become extra smooth [35,36].

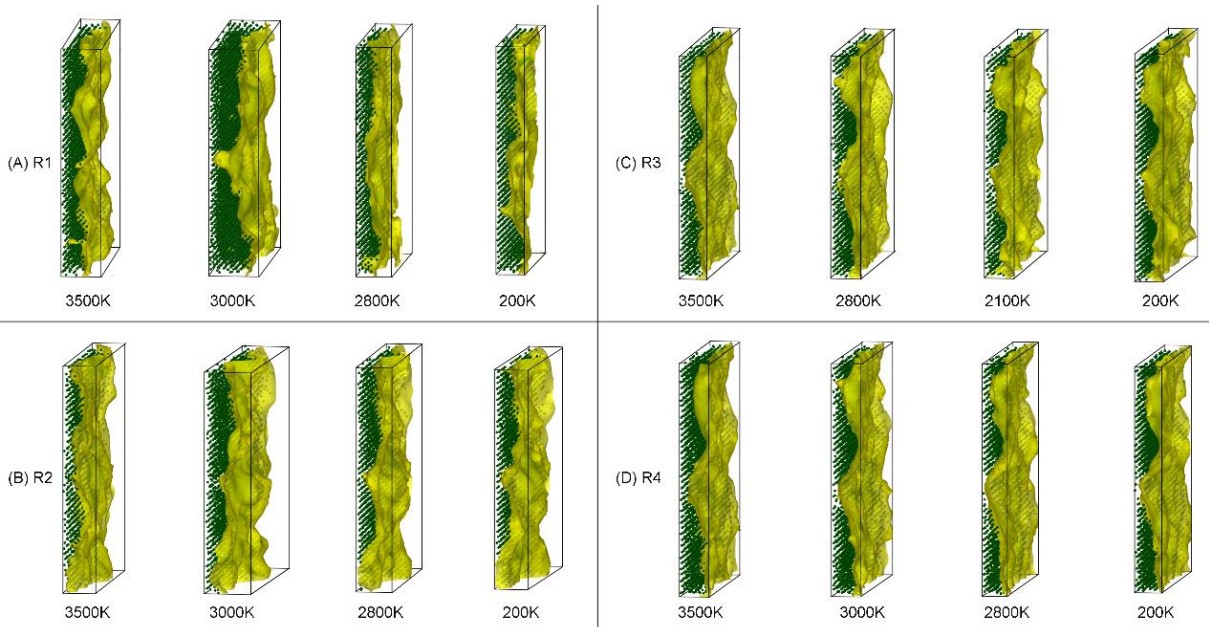

**Figure 10.** Evolution of solid–liquid interface slices at different cooling rates, R1: $10^{10}$ K/s, R2: $10^{11}$ K/s, R3: $10^{12}$ K/s, R4: $10^{13}$ K/s.

## 4. Conclusions

Based on the presented analysis regarding the evolution properties of structural defects in the SiC solid–liquid interface during rapid cooling, the following conclusions can be obtained.

At lower cooling rates, various disordered polyhedral structures are gradually transformed into ordered structures, forming crystal structures, and Voronoi polyhedron transform into < 4 0 0 0 >. With the increase in time, there are more and more ordered structures in the system.

Upon quenching, the number of defects in the system decreases, and the distribution is concentrated at the solid–liquid interface. The change of the number of crystal and its neighboring atoms directly reflects the degree of crystallization and the number of defective structures contained in the system.

For a cooling rate of $10^{10}$ K/s, the energy of the system changes. At higher cooling rates, the energy change of the system is not obvious, indicating that the system has not fully crystallized. The interfacial evolution process of the solid–liquid phase transformation proves that the surface morphology of the interface greatly affects the crystal quality and crystallization rate.

**Author Contributions:** Conceptualization, X.G. and Y.G.; funding acquisition, X.G. and T.G; investigation, X.G. and Y.G.; methodology, Y.G. and T.G.; software, X.G. and Y.G.; validation, T.G.; writing—original draft preparation, Y.G. and Z.M.; writing—review and editing, X.G. and T.G.; supervision, X.G. and T.G. All authors have read and agreed to the published version of the manuscript.

**Funding:** This work was supported by the National Natural Science Foundation of China (Grant nos.51761004, 51661005, and 11964005), the Guizhou Province Science and Technology Fund (Grant no. ZK [2021] 051, [2017] 5788, and J [2015] 2050), High level Creative Talent in Guizhou Education Department of China, and the Cooperation Project of Science and Technology of Guizhou Province (Grant no. LH [2016] 7430).

**Data Availability Statement:** Not applicable.

**Conflicts of Interest:** The authors declare that they have no conflict of interest.

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
