# Peer review of "Effect of Cooling Rate on the Crystal Quality and Crystallization Rate of SiC during Rapid Solidification Based on the Solid–Liquid Model"

_crystals, doi:10.3390/cryst12081019_

Round 1

Reviewer 1 Report

This paper is presented MD simulation results for a better understanding of crystallization and its rate. The results clearly show what happened during the initial stage at the solid-liquid interface and spontaneously changed by cooling speed in MD simulations. And finally the authors successfully presented that the solid-liquid interface transformation directly affects the surface morphology, crystal quality and crystallization rate.

I understand MD simulation is not a general research topic and not so many people work on it. Therefore this unique report would be useful.

Reviewer 2 Report

The authors investigated the quenching process of SiC liquid by MD simulation.I think that a certain amount of knowledge has been obtained but  corrections are necessary for the publication of the publication.

(1) About the importance of the quenching process

  The first paragraph of the introduction mainly explains the characteristics of SiC crystals. But I think that the crystals with less defects are suitable to obtain any of such characteristics.

  On the other hand, the cooling rate of above 10 ^ 10K / s considered here is a kind of calculation method which achieves amorphization,

and is practically impossible and unsuitable to obtain the above propertie.

So I did not understand the significance of performing MD calculation in this paper. So I would like the authors to explain it.

(2) About SiC liquid phase

As far as the reviewers know, SiC does not form a liquid phase under normal pressure.

Please describe how to assume the generation even in the calculation, including the practicality of this calculation.

(3) Which part did the authors obtaine RDF in Fig.2 ? The entire right half of the cell in Fig.1 ?

Although it is described that it tends to crystallize at a cooling rate of 10 ^ 10K,

did the authors compare with RDF from  the crystal on the left side?

I wonder whether the authors found the clustering or the crystalization.

(4) The authors describe cubic diamonds in Figure 3 and Figure 8, but I don't know what they are.

Is it a 4-coordinated tetrahedral structure centered on C or Si?

(5)  I did not understand the importance of Figure 6, but does it show the main four-coordination structure and its increase with temperature decreases?

Tersoff potential explains the covalent bonding so I think that the 4-coordination structure should be the main one, so what is the intention?

(6)The information in Figure 7 is included in Figure 9, so please summarize it.

(o) Just an opinion for reference.

I understand that SiC crystallizes even in the quenching process, but it seems not a remarkable result other than that.

As a researcher dealing with the growth of SiC, I think it would be useful to discuss the uptake and diffusion at the interface in this process.

Reviewer 3 Report

In this manuscript, the authors present molecular dynamic simulation for solidification of 3C-SiC from SiC liquid by quenching. A critical cooling rate of 10e10 K/s is preferable for crystallization, whereas disordered structure is formed at higher cooling rates. This manuscript fits the scope of the journal and is considerable for publicization after solving the below issues.   1. Some typos. e.g., P1L36: At the same time is underlined, P7L201: rateswere, etc. 2. Please define abbreviations before use, such as PRDF (Pair Radial Distribution Function), MSD, etc. 3. P7L195: "This result indicates that a lower cooling rate is conducive to the transformation of the high coordination structure to the <4 0 0 0> structure, giving the system more time for the atom positions to be adjusted."  P10L282: "At low cooling rate, various disordered polyhedral structures are gradually transformed into ordered structures, and crystal structures are formed <2 3 0 0> voronoi polyhedron is not converted to <4 0 0 0>, so the system does not produce new crystals." Those two statements seem to conflict with each other. Please clarify them.

Round 2

Reviewer 2 Report

In the Authors response, please specify whether the authors have modified the text or not. Responses that require reviewers to check all points again are not good.
Also, in the reply to 1st question, does 10 ^ 9K / s cooling rate suggest that 3x10 ^ -6 s from 3000K to 0 K ? The answer of 17 days is not very appropriate.

Author Response

Comments and Suggestions for Authors

In the Authors response, please specify whether the authors have modified the text or not. Responses that require reviewers to check all points again are not good.

 Also, in the reply to 1st question, does 10 ^ 9K / s cooling rate suggest that 3x10 ^ -6 s from 3000K to 0 K? The answer of 17 days is not very appropriate.

 Response: Thank you for your comments.

We have marked the corrections in the text content in green font, mainly for the corrections of issues raised by your reviewers, grammatical errors, and incoherence.

The determination of the cooling rate is related to the configuration of the server, which greatly affects how fast or slows the calculation is.

The time step we set is 1fs and the temperature is from 3500K to 200K, a total of 330,000,000 steps. The cooling rate under this condition is 1010 K/s. We have a computational server with 24 cores in one node, and we use 4 nodes in this model.  Our model is 64,000 atoms, and when testing the computational efficiency, our server was able to run 13,000 steps per minute, and the cooling rate of 1010 K/s was 330 million steps in total, taking 25,384 minutes or 423 hours. It takes 17 days to calculate the cooling rate of 1010 K/s (330ns), therefore we want to get a lower cooling rate and we need more computational resources.

In the study of molecular dynamics simulations of rapid quenching processes, the cold rate is a comparably important variable, and 1010 K/s is a relatively low cooling rate that is informative for the study of condensed matter [1-4].

[1] Luo J, Gao T, Ren L, et al. Segregation phenomena of As in GaAs at different cooling rates during solidification[J]. Materials Science in Semiconductor Processing, 2019, 104: 104680.

[2] Celik F A. Molecular dynamics simulation of polyhedron analysis of Cu–Ag alloy under rapid quenching conditions[J]. Physics Letters A, 2014, 378(30-31): 2151-2156.

[3] Papanikolaou M, Salonitis K, Jolly M, et al. Large-scale molecular dynamics simulations of homogeneous nucleation of pure aluminium[J]. Metals, 2019, 9(11): 1217.

[4] Deng L, Du J. Effects of system size and cooling rate on the structure and properties of sodium borosilicate glasses from molecular dynamics simulations[J]. The Journal of Chemical Physics, 2018, 148(2): 024504.

We have also made changes in the manuscript to respond to the issues raised in the first round of review concerning the content of the figures. We have also responded in detail to your questions, and this process has enabled us to sort out the logic and details once again. Thank you.
